# MTOR Signaling and Metabolism in Early T Cell Development

**DOI:** 10.3390/genes12050728

**Published:** 2021-05-13

**Authors:** Guy Werlen, Ritika Jain, Estela Jacinto

**Affiliations:** Department of Biochemistry and Molecular Biology, Robert Wood Johnson Medical School, Rutgers University, Piscataway, NJ 08854, USA; guy.werlen@rutgers.edu (G.W.); rj420@scarletmail.rutgers.edu (R.J.)

**Keywords:** mTOR, mTORC1, mTORC2, thymocytes, T lymphocytes, early T cell development, T-cell metabolism

## Abstract

The mechanistic target of rapamycin (mTOR) controls cell fate and responses via its functions in regulating metabolism. Its role in controlling immunity was unraveled by early studies on the immunosuppressive properties of rapamycin. Recent studies have provided insights on how metabolic reprogramming and mTOR signaling impact peripheral T cell activation and fate. The contribution of mTOR and metabolism during early T-cell development in the thymus is also emerging and is the subject of this review. Two major T lineages with distinct immune functions and peripheral homing organs diverge during early thymic development; the αβ- and γδ-T cells, which are defined by their respective TCR subunits. Thymic T-regulatory cells, which have immunosuppressive functions, also develop in the thymus from positively selected αβ-T cells. Here, we review recent findings on how the two mTOR protein complexes, mTORC1 and mTORC2, and the signaling molecules involved in the mTOR pathway are involved in thymocyte differentiation. We discuss emerging views on how metabolic remodeling impacts early T cell development and how this can be mediated via mTOR signaling.

## 1. Introduction

T cells originate from hematopoietic stem cells that home into the thymus and differentiate into two main T cell lineages with distinct immune functions. The γδ-T cells will egress the thymus as functional effector cells and migrate to the epidermis, mucosa and intestine. However, most T cells developing in the thymus belong to the αβ-lineage. They differentiate further into either naïve T-helper, T-cytotoxic or T-regulatory subsets that exit the thymus to settle into peripheral organs such as spleen and lymph nodes. These naïve T cells that express a diverse repertoire of αβ-T cell receptors (αβ-TCR) are poised to become activated upon encounter with their cognate antigen. Signals from the αβ-TCR, coreceptors and cytokines trigger a cascade of intracellular events culminating in changes in gene expression that directs cell fate [1,2]. Recent studies on peripheral T cell activation along with cancer metabolism have unraveled how nutrient availability and metabolism impact cell signaling and vice versa [3,4]. While the role of metabolic reprogramming in the activation of peripheral T cells has been relatively well-delineated, its role in early T cell development in the thymus is just surfacing. Studies on immune cell metabolism have been facilitated by the use of immunosuppressants such as rapamycin, which targets mTOR, a central controller of metabolism. In this review, we will focus on the involvement of mTOR signaling and metabolism in early T cell development in the thymus.

## 2. Rapamycin and Antimetabolites as Immunosuppressants

Rapamycin (sirolimus), originally isolated as an antifungal antibiotic from the filamentous bacteria, *Streptomyces hygroscopicus* [5], has potent immunosuppressive and anti-tumor properties [6]. In rats, pioneering studies reveal that it prevents the development of experimental immunopathies, including experimental allergic encephalomyelitis (EAE) and adjuvant arthritis (AA) and the generation of a humoral antibody [7,8]. The treatment of animals with rapamycin during organ transplantation revealed immunosuppression with less toxicity compared to other immunosuppressants such as cyclosporine A (CsA) and FK506, as well as improved graft survival [9]. Rapamycin also has synergistic action together with other immunosuppressants, resulting in less toxicity. Many clinical trials supported its benefits during organ transplantation; hence, its use as an immunosuppressant to prevent allograft rejection was approved in 1999 [10].

Rapamycin forms a complex with the prolyl isomerase FKBP12, and together, they bind to the FKBP12-rapamycin binding region (FRB) in mTOR [11]. Early studies to elucidate the immunosuppressive properties of rapamycin revealed that it blocks T cell proliferation that is induced by interleukin-2 (IL-2) or IL-4 [12]. It prevents G1 to S transition of the cell cycle in T cells stimulated with IL-2 [13]. The identification of the cellular target of rapamycin (TOR) facilitated the understanding of not just how rapamycin confers immunosuppression but uncovered how cells control cell growth and metabolism [14,15].

Although studies on the role of metabolism in T cell development and activation has undergone a renaissance only in the past years, blocking metabolic pathways through the use of antimetabolites have been in use for decades. Azathioprine, a nucleotide-blocking agent that prevents DNA synthesis, has been used in solid organ transplantation since the 1960s [16]. Methotrexate, an antifolate, inhibits dihydrofolate reductase, thus diminishing the generation of thymidine, leading to impaired DNA synthesis. Although originally used as an anti-cancer drug, methotrexate is effectively used in autoimmune disorders and organ allografts since the 1960s [17,18]. Mizoribine, an inhibitor of inosine monophosphate (IMP) synthetase and guanosine monophosphate synthetase, blocks guanine nucleotide synthesis. It is also used as an immunosuppressant to prevent organ allograft rejection in humans and animal models [19,20]. Mycophenolates are converted to mycophenolic acid in the liver, and this metabolite inhibits the rate-limiting enzyme, inosine monophosphate dehydrogenase, of the de novo synthesis of guanosine nucleotides [21]. T-cell proliferation is highly dependent on the de novo synthesis of nucleotides. The efficacy of these anti-metabolites likely relies on the principle that highly proliferating cells, such as an activated T cell, are sensitive to the limitation of crucial metabolites or nutrients that are required for macromolecule biosynthesis.

## 3. mTOR Signaling

The target of rapamycin (TOR) was first identified in yeast through genetic experiments. The identification of mammalian TOR (mTOR; later renamed mechanistic target of rapamycin) soon followed, revealing that TOR/mTOR, a serine/threonine protein kinase, is a conserved protein kinase from yeast to man [22].**** mTOR forms two distinct protein complexes: mTOR complex 1 and 2 (mTORC1 and mTORC2) (Figure 1). mTORC1 consists of mTOR, raptor, and mLST8 [23,24]. Other, less-conserved proteins also bind to mTORC1, including PRAS40 and Deptor. mTORC2 contains mTOR, Rictor, SIN1, and mLST8. In higher organisms, mTORC2 also associates with Protor and Deptor. Although the catalytic activity of these protein complexes is provided by mTOR, their unique composition, as well as the regulation of the post-translational modification of their distinct components, confer functional specificity to mTORCs. In addition, their cellular localization also accounts for target specificity. While both mTORCs localize to the membrane periphery, mTORC1 is recruited to the periphery of lysosomes and Golgi, whereas mTORC2 has been shown to localize to plasma membrane, endosomes and to the mitochondria-associated endoplasmic reticulum membrane (MAM) [25,26].

Since mTOR signaling has been extensively reviewed elsewhere [23,24,27], here, we will only give a general overview of mTORC1 and mTORC2 signaling.

### 3.1. mTORC1

mTORC1 integrates signals from a variety of environmental cues, including nutrients and growth factors/hormones, as well as stress inputs in order to maintain metabolic homeostasis. It orchestrates anabolic metabolism by regulating key effectors of different biosynthetic and metabolic processes while repressing catabolic processes such as autophagy. The most well-appreciated mTORC1 function is its role in modulating protein synthesis in response to amino acid levels. A hallmark of mTORC1 activation is the phosphorylation of translation regulators, S6K1 and 4E-BP1. The mTORC1 component, raptor, presents substrates to mTOR as part of this complex. Changes in the post-translational modifications or expression levels of raptor modulate mTORC1 activity [23]. mTORC1 localizes to different cellular compartments to perform its functions, but the most well-characterized is its localization to the periphery of lysosomes. Lysosomes recycle macromolecules by acting as a storage unit for intracellular nutrients. The localization of mTORC1 in this compartment underscores how this complex is activated by the availability of nutrients such as amino acids. The activation of mTORC1 in lysosomes is mediated by Rags (Ras-related GTP binding proteins) [28,29,30]. In the presence of amino acids, RagA/B binds to GTP while Rag C/D binds to GDP. The Rag heterodimers interact with raptor to modulate the translocation of mTORC1 to the surface of the lysosomes. Amino acid availability is communicated to mTORC1 via distinct amino acid “sensors” that then modulate Rag signaling [31]. There are also some studies that demonstrate how other nutrients/metabolites could more directly modulate mTORC1 [32,33]. Whether there are other sensors that convey nutrient availability to mTORC1 would need further investigation. Under nutrient-abundant conditions, mTORC1 also inhibits autophagy by the negative regulation of key mediators of this process, including Atg13 and ULK1 [34,35,36]. Thus, the activation of mTORC1 by the presence of nutrients positively regulates growth and anabolic processes while it negatively regulates catabolic processes under normal/non-pathological conditions.

The activity of mTORC1 is also modulated during increased PI3K (phosphatidylinositol-3 kinase) signaling. PI3K activation is triggered by receptor tyrosine kinase (RTK) signaling. In the case of T cells, TCR/CD3 and cytokine receptor signaling enhance PI3K activity. Activated PI3K phosphorylates the inositol ring of PIP2 (phosphatidyl-inositol-4,5-bisphosphate), a membrane phospholipid, and converts it to PIP3 (phosphatidylinositol 3,4,5-trisphosphate) at the plasma membrane. Following the generation of PIP3, a subset of signaling proteins like Akt and PDK1 is recruited to the membrane due to the affinity of their pleckstrin homology (PH) domain for PIP3.

Increased PI3K signaling downregulates the activity of the tuberous sclerosis complex TSC (composed of TSC1, TSC2 and TBC1D7), a negative regulator of mTORC1 [37,38,39]. Rheb, another GTPase, is modulated by TSC and binds to mTORC1 to increase its catalytic activity [40]. The increased PI3K signaling enhances the catalytic activity of several AGC (Protein Kinase A, PKG and PKC) kinase family members, including S6K and Akt, by promoting the phosphorylation of their catalytic loop by PDK1, another AGC kinase [41]. The Akt-mediated phosphorylation of TSC2 downregulates TSC activity, thus relieving its suppression of mTORC1. PDK1 also phosphorylates S6K1 to augment its catalytic activity. mTORC1 further enhances S6K1 activity by phosphorylating its hydrophobic motif (HM), an allosteric site that is conserved among AGC kinases. Other protein kinases such as MAPK and RSK, which are activated by mitogenic signals, also enhance mTORC1 activity by modulating the phosphorylation of TSC [42,43]. Unfavorable growth conditions, such as during nutrient limitation or other stress stimuli, dampen mTORC1 activity via TSC. During energy deprivation, AMPK (AMP-activated protein kinase) phosphorylates and enhances TSC2 activity, which leads to mTORC1 downregulation [37]. This is critical, as it helps prevent apoptosis as a result of energy depletion. Another negative regulator of mTORC1 signaling is PRAS40 (proline-rich Akt substrate of 40 kDa). The binding of PRAS40 to raptor inhibits mTORC1, while its phosphorylation by Akt and mTOR dissociates PRAS40 from the complex during favorable growth conditions [44,45,46].

mTORC1 is modulated by post-translational modifications of its components mTOR and raptor. mTOR is phosphorylated at several sites that positively regulate its activity in response to the presence of growth signals [47,48,49,50]. Raptor has multiple phosphorylation sites that are targeted by various protein kinases, leading to either positive or negative regulation of mTORC1 activity.

### 3.2. mTORC2

Compared to mTORC1, the regulation of mTORC2 is far less well-understood. However, a hallmark of mTORC2 activation is the phosphorylation of Akt at its allosteric sites—namely, Ser473 at the hydrophobic motif (HM) and Thr450 at the turn motif (TM). The phosphorylation of Akt at Ser473 is induced by PI3K signaling, such as during stimulation with growth factors. The Akt HM phosphorylation is also enhanced during nutrient limitation or nutrient restimulation [51,52]. Hence, unlike mTORC1 that is activated by increased nutrient levels, the activation of mTORC2 seems to respond to stress conditions or destabilization of metabolic homeostasis. mTORC2 activity is also linked to non-PI3K-dependent mechanisms. mTORC2 targets such as Akt Thr450 and PKC TM and HM phosphorylation occur constitutively, and whether they are responsive to nutrient levels remains to be further examined [53,54,55,56].

Changes in posttranslational modifications of the mTORC2 components, mTOR, rictor and SIN1 in response to growth signals modulate mTORC2 activity. mTOR autophosphorylation at Ser2481 is normally detected in both mTORC1 and mTORC2, but this phosphorylation is absent in mTORC2-disrupted cells [55,57,58,59]. In response to glucose limitation, mTORC2 is activated, and mTOR is phosphorylated via AMPK [51,52]. Rictor is also phosphorylated at many sites, mostly at the carboxyl-terminal region. While the functions of these phosphosites remain to be characterized, so far, most of the ones that have been studied appear to negatively regulate mTORC2 activity [60,61,62]. SIN1 also undergoes phosphorylation to modulate mTORC2 activity. The phosphorylation at Thr86 of SIN1 occurs via Akt, but whether this phosphorylation positively or negatively impacts mTORC2 activity is controversial [63,64]. Other phosphosites on SIN1 have been recently identified, but their functions remain to be further investigated [65].

mTORC1 signaling also negatively feeds back onto growth factor/PI3K signaling and, subsequently, dampens mTORC2 activity. During enhanced mTORC1 signaling, increased S6K1 activity promotes phosphorylation of the insulin receptor substrate (IRS-1), leading to the downregulation of PI3K/IRS-1 signaling [66]. However, this can be reversed with the use of rapamycin, which decreases S6K1 activity, leading to enhanced PI3K signaling. Furthermore, the presence of rapamycin upregulates the expression of growth factor receptors, which also causes an increase in PI3K signaling and, thus, mTORC2 activity [67]. This feedback signaling indicates that, as the nutrient levels and mTORC1 activity decline, increased mTORC2 signaling could upregulate mechanisms that provide intracellular nutrients and, thus, maintain cell survival.

## 4. Overview of mTOR Functions in Cell Metabolism

Both of the mTOR complexes are involved in the control of cellular metabolic processes (Figure 2) (reviewed in [23,24,27]). mTORC1, which is activated in the presence of sufficient nutrients, promotes anabolic metabolism. Emerging studies also link mTORC2 to the control of the metabolic processes. Interestingly, mTORC2 responds to nutrient fluctuations and could thus have functions in both anabolic and catabolic metabolism. The overlapping and distinct functions of the mTORCs in metabolism remain to be delineated.

mTORC1 mainly controls various aspects of anabolic metabolism [23]. mTORC1 promotes glucose metabolism via modulation of the key transcription factors, HIF1α and Myc, that control the expression of the genes involved in glycolysis and the TCA cycle [68]. Highly proliferating cells such as activated T cells and cancer cells have enhanced glucose consumption and metabolism, corresponding to augmented mTORC1 activity [69]. Those cells upregulate, in an mTORC1-dependent manner, critical metabolic enzymes of glycolysis such as hexokinase, which phosphorylates glucose in the initial step of glycolysis and pyruvate kinase M2 (PKM2), an isoform that is expressed during increased aerobic glycolysis. Increased mTORC1 activity also enhances the phosphofructokinase-2/fructose-2,6-bisphosphatase B3 isotype (PFKFB3) via HIF1α [70]. mTORC1 also modulates lactate dehydrogenase, a tetrameric enzyme that converts pyruvate to lactate during aerobic glycolysis of highly proliferating cells (Warburg effect) [71]. Various aspects of the TCA cycle, oxidative phosphorylation (OxPhos) and mitochondrial biogenesis are also controlled by mTORC1 [72,73,74]. mTORC1 modulates the expression of genes involved in mitochondrial functions. The inhibition of mTORC1 using rapamycin reduces mitochondrial membrane potential, oxygen consumption and ATP synthetic capacity [75].

The upregulation of glycolysis and the TCA cycle in highly proliferating cells does not only produce more ATP, but also provides intermediates that are utilized for the synthesis of cellular building blocks, such as amino acids, lipids, and nucleotides. mTORC1 is involved in the regulation of key metabolic enzymes and transcription factors that are required for these biosynthetic pathways. For example, mTORC1 promotes lipogenesis by modulating the key lipogenic transcriptional regulator, SREBP (sterol regulatory element binding protein), which, in turn, controls the expression of lipogenic enzymes such as ACC, FASN and SCD1 [76]. mTORC1 also negatively regulates lipid catabolism. When mTORC1 is inhibited or disrupted in the adipose tissue, lipolysis is enhanced by the adipose triglyceride lipase (ATGL), while fatty acid oxidation (FAO) is increased in part due to the increased activity of carnitine palmitoyltransferase (CPT1), which regulates the FAO pathway. mTORC1 has multifaceted roles in the control of protein synthesis [77]. It addition to phosphorylating the translation regulators, S6K and 4E-BP1, it also controls the biogenesis of ribosomes, the molecular engines of mRNA translation. mTORC1 positively regulates rRNA transcription, as well as the generation of ribosomal proteins and their assembly factors [78,79]. Highly proliferating cells have also increased need for nucleotides, which are utilized for DNA and RNA synthesis, as well as for ribosome biogenesis. mTORC1 regulates the de novo synthesis of pyrimidines via the activation of the rate-limiting enzyme CAD (carbamoyl-phosphate synthetase 2, aspartate transcarbamylase and dihydro-orotase) [80,81]. mTORC1 also augments flux through de novo purine synthesis via modulating the transcription of enzymes involved in this biosynthetic pathway [82]. Hence, mTORC1 orchestrates anabolism by regulating various metabolic pathways at different levels. While we have a better understanding of how mTORC1 is activated by amino acids to promote protein synthesis, there is little known on how it responds to other nutrients in order to conduct its functions in anabolic metabolism.

Although studies to unravel the functions of mTORC2 in metabolism have lagged behind, there is accumulating evidence supporting the role of this complex in modulating key transcription factors and anabolic enzymes. The mTORC2 substrate, Akt, positively regulates glucose metabolism via multiple mechanisms. Furthermore, knockdown and knockout models have also revealed that mTORC2 controls lipogenesis both in an Akt-dependent and -independent manner [83]. mTORC2 also functions in protein synthesis via the modulation of ribosome biogenesis and amino acid transport [84,85]. It is also involved in the processing of nascent peptides [53,54]. By modulating hexosamine biosynthesis, which produces critical metabolites for glycosylation, mTORC2 also regulates protein folding [51]. mTORC2, via Akt, is also linked to the regulation of the pentose phosphate pathway, as well as to purine synthesis [86,87,88]. Unlike mTORC1, which is activated in the presence of nutrients to promote anabolic metabolism, mTORC2 seems to respond to nutrient fluctuations instead, suggesting that it functions in establishing metabolic homeostasis.

The activation of naïve αβ-T cells upon an encounter with antigen-presenting cells in the periphery provides the best example of how metabolic reprogramming shapes cellular responses [89]. Prior to activation, naïve αβ-T cells have low rates of nutrient uptake and rely on oxidative metabolism to maintain survival. Upon activation, they increase nutrient uptake, enhance the glycolytic flux through biosynthetic pathways in order to generate the energy and intermediates necessary to synthesize macromolecules for growth and proliferation. A number of these processes have been shown to be dependent on mTOR signaling [90,91]. The mTORC1-dependent metabolic reprogramming that takes place upon T-cell activation is also partly mediated by Myc [92]. After differentiation to effector cells and the subsequent expansion that allows T cells to mount a robust immune response, the surviving T memory cells switch back to a naïve T cell-like oxidative metabolism wherein mTORC1 signaling becomes downregulated [93]. The peripheral Tregs, which have immunosuppressive functions, also have different metabolic needs than effector cells [94]. Tregs are dependent on cholesterol and lipid metabolism; in particular, they upregulate the mevalonate pathway in an mTORC1-dependent manner to sustain Treg proliferation and suppressive function. The role of the mTOR complexes in peripheral T cell activation and metabolic reprogramming has been extensively covered by previous reviews [3,69,91].

## 5. Early T Cell Development in the Thymus

All T cells originate from bone marrow-derived hematopoietic stem cells (HSCs) that migrate into the thymus at this organ’s corticomedullary junction. In the thymus, signaling through the Notch family of receptors differentiate HSCs into thymocytes (the immature precursors of naive T cells) and commits them to the T cell lineage. Two major T lineages emerge from thymocyte maturation: the αβ- and γδ-T cells defined by their respective expression of αβ- or γδ-T cell receptor (TCR) complexes (Figure 3) [95,96]. Lineage immature thymocytes, characterized as CD4^-^CD8^-^ double-negative 1 (DN1) and “lineage negative” [NK1.1^−^ (NK marker), B220^−^ (B cell marker), TER119^−^ (erythroid marker), Gr1^−^ (neutrophil marker) and CD3ε^−^ (T cell marker)], express CD44, cKit and CD24 but not CD25. The induction of CD25 expression marks the entry into the double-negative 2 (DN2; CD4^-^CD8^-^) stage of thymocyte development. At DN2, the expression of recombination-activating genes Rag1 and Rag2 drives the recombination of the V(D)J segments of the β-, γ- or δ-TCR gene loci and thymocyte maturation continues into the double-negative 3 (DN3; CD4^-^CD8^-^) stage. At DN3, thymocytes commit irreversibly to the T cell lineage, which is characterized by alterations in the ratio of specific transcription factor family members Runx3/Runx1, Gfi1B/Gfi1 and PU.1/SpiB, indicating changes in the transcriptional network. Similarly, distinct chromatin landscape and accessibility account in part for the divergence of γδ- and αβ-T cell lineages [97].

The emergence of the γδ-lineage that commences at DN2 by the distinct recombination of the TCR subunit genes *Tcrd* and *Tcrg* is believed to be complete by the DN3 stage [98,99]. Thymocytes that express a functional γδ-TCR have chiefly accomplished their development in the thymus and translocate to their peripheral locations as discussed in Section 5.2. In contrast, the successful rearrangement of *Tcrb* genes at the DN2 and early DN3 (DN3a) stages commits developing thymocytes to the αβ-lineage. Signals from the pre-TCR (TCRβ in complex with invariant pre-TCRα and CD3 signaling molecules) and Notch rescue thymocytes from apoptosis and enforce TCR-β selection by mediating allelic exclusion at the *Tcrb* gene locus. Notch, along with pre-TCR signals, promote the proliferation of β-selected thymocytes [100]. Thus, the DN3b stage is marked by productive TCRβ selection that promotes cell cycle progression and triggers cell proliferation along the αβ-lineage [100]. Notch signaling involves engagement of the Notch receptor (Notch1) with its ligand Delta-like ligand 4 (DLL4), leading to proteolytic cleavage of Notch1 wherein its intracellular domain translocates to the nucleus and acts as a transcriptional regulator. It is noteworthy that Notch target genes include the pre-TCRα gene, *Ptcra*. Notch signals are crucial for the survival, proliferation and differentiation of αβ-lineage DN thymocytes up to the DP stage. The high expression of Notch in the DN stage is consistent with its role in pre-T cell survival and efficient β-selection. In addition to pre-TCR and Notch signals, IL-7 maintains the survival of DN thymocytes by promoting the expression of the pro-survival modulator Bcl-2 [101]. It also upregulates genes involved in cell growth, while suppressing the expression of the transcriptional repressor Bcl-6 in DN3 and DN4 thymocytes. Pre-TCR signaling allows DN3a to DN3b transition followed by differentiation into the Kit^-^CD44^-^CD25^-^ DN4 stage. At DN4, thymocytes may commence the expression of CD8 and are referred to as immature single positive (CD8-ISP) cells. CD8-ISP is not just a transitional stage but has a distinct gene expression profile from the DN and DP stages [102]. After a single round of the cell cycle, ISP cells start expressing both the CD8 and CD4 coreceptors and, thus, progress to the CD4^+^CD8^+^ double-positive (DP) stage. The transcription factors TCF-1, LEF-1 and RORγt are among roughly 40% of genes that change their expression profile during the CD8-ISP to DP transition. Notch signaling is blocked by pre-TCR expression and could thus facilitate the transcriptional changes associated with the DP stage [103]. At the DP stage, gene rearrangement of TCRα ensues, leading to the assembly of immunologically functional αβ-TCR.

DP thymocytes sample the thymic cortical epithelial cells (TEC) for the possible binding of their TCR with major histocompatibility complex (MHC) that are expressed on the surface of TECs and that are loaded with self-peptides. The inability to bind self-peptide/MHC leads to apoptosis, whereas DP thymocytes that are capable of interacting with MHC molecules are positively selected, and their maturation to either CD4^+^ or CD8^+^ single-positive (SP) cells ensues. If the affinity for self-peptide/MHCs is too strong, they either undergo apoptosis (negative selection) or are diverted to other lineages, such as Foxp3^+^ T-regulatory cells (Treg). However, only a small subset of αβ-ΤCR-expressing cells differentiate into Treg, as well as natural killer T cells (NKT), in the thymus. The process of negative selection and the generation of Tregs are critical for self-tolerance to prevent autoimmunity. αβ-thymocytes that successfully go through these processes and reach the SP stage subsequently exit the thymus and home into peripheral lymphoid organs as naïve T-helper cells expressing the CD4 coreceptor and CD8-expressing naïve cytotoxic T cells or as functional Tregs. In these compartments, naïve T cells are braced to recognize the corresponding antigens presented by antigen-presenting cells (APCs) and differentiate into either effector cells ready to mount an acute immune response or memory T cells, the guardians of future immune intervention.

In addition to signals derived from the pre-TCR and other cell surface receptors, such as Notch, that regulate αβ-T cell development, evidence is accumulating that nutrient metabolism plays a significant role in thymocyte maturation. Early studies utilizing unsorted, heterogeneous thymocytes unraveled that the oxidative phosphorylation (OxPhos) of glucose was the main source of energy in resting thymocytes, while the glycolytic degradation of glucose to lactate was enhanced in proliferating thymocytes following mitogen stimulation [104]. More recent studies have demonstrated that DN3 and DP thymocytes favor OxPhos over glycolysis, whereas DN4 and ISP cells upregulate both OxPhos and glycolysis [105,106]. DN3 thymocytes have low rates of glucose and glutamine consumption, whereas DN4 cells that are undergoing rapid proliferation upregulate the uptake of these nutrients [107]. The increased expression of distinct nutrient transporters, such as the transferrin receptor CD71, which mediates iron uptake, CD98, which is involved in glutamine transport, and the phosphate transporter also underscores how nutrient availability contribute to thymocyte development [108,109]. DN4 and ISP cells demonstrate a higher expression of CD98 and CD71 [105]. They also have a larger cell size and S6 phosphorylation, supporting increased protein synthesis. Highly proliferating DN thymocytes are also sensitive to Glut1 deficiency [110]. The increased expression of Glut1 at the DN3 to DN4 stages, the most proliferative phases of thymopoiesis, are consistent with the sensitivity of these subsets to Glut1 deficiency. Furthermore, these findings support the notion that there is enhanced glucose metabolism during these stages of thymocyte differentiation. Transcriptional changes during the DP stage—in particular, the small DP stage, which is transitioned from the DP blast stage—leads to a reduction in the genes associated with metabolic and housekeeping activities [103]. The passage from small DP to CD69^+^-DP (characterizes activated DP subsets following TCR engagement by peptide/MHC) undergo a dramatic remodeling of their transcriptome. Among the induced genes are those involved in the biosynthetic and oxidative pathways. The pyruvate dehydrogenase complex kinases Pdk1 and Pdk2 are downregulated in CD69^+^-DP, while the upstream enzymes of the glycolytic pathway are upregulated. Cellular housekeeping genes, including those involved in increased protein synthesis, are also reactivated. Thymic SP cells have a more mitochondrial content, as well as higher ROS and mTORC1 activation, compared to recently emigrated CD4^+^- cells [111]. These findings suggest that thymic CD4-SP cells are more metabolically active and that the metabolic quiescence of peripheral naïve T cells is achieved after the egress from the thymus. The metabolic requirements of other thymocyte subsets has also begun to be investigated.

### 5.1. αβ-T Cells

Studies using rapamycin provided initial clues on the role of mTOR in T cell development in the thymus. In mice, rapamycin treatment leads to thymic atrophy due to diminished proliferation without increasing apoptosis [112,113]. Rapamycin treatment in vivo blocks the DN3 to DP differentiation [114]. However, it does not affect negative selection [112]. In rats, thymic atrophy also occurs but recovers after a few weeks upon cessation of rapamycin treatment [115]. Rat DP thymocytes, but not the DN and SP subsets, have accelerated apoptosis [116]. While these studies unraveled the involvement of mTOR in early thymocyte development, they do not distinguish whether the effects are due to inhibition of mTORC1 and/or mTORC2 signaling, since rapamycin is an allosteric inhibitor that only blocks some of mTORC1 functions. Furthermore, prolonged rapamycin treatment may also disrupt mTORC2 assembly [117]. Hence, to delineate the specific functions of mTORC1 and mTORC2 in thymocyte development, genetic experiments that ablate mTORC components and their downstream effectors have provided better insights on the specific role of mTORC1 and mTORC2 signaling, as discussed below (Figure 4).

#### 5.1.1. The Role of mTORC1 in αβ-T-Cell Development

The expression of raptor, a unique component of mTORC1 but not mTORC2, has been knocked out specifically in thymocytes, and the results support a role for mTORC1 during early T cell development in the thymus. Thymic atrophy was observed using *CD2-Cre* to abrogate *Raptor* at the DN1 stage or the tamoxifen inducible *Rosa26-CreERT2* system to ablate it in all tissues, including hematopoietic cells [105,114,118]. In both genetic knockout systems, the proportion of DN1 cells is increased, whereas the proportion and absolute numbers of DN2 and DN3 subsets are reduced, indicating a block at the early stage of thymic differentiation [114]. DP cells have decreased percentage and absolute number, and the proportion of CD4^+^^+^ and CD8^+^^+^ SP cells is also diminished. However, although rapamycin treatment and raptor deficiency both result in thymic atrophy, their targeted thymocyte subsets are different. Rapamycin blocks the DN3 to DP transition, whereas raptor deficiency induces defects from the very early DN1 stage up to the late stages of thymocyte ontogeny, indicating a role for mTORC1 at different developmental steps [114]. Raptor deficiency promotes cell cycle defects with instability of the cyclin D2/D3-CDK6 complexes in early T cell progenitors [114]. Thus, raptor-deficient DN1 cells have decreased S/G2/M phases and an increased G0/G1 phase, supporting that mTORC1 promotes proliferation. An in vitro culture with stromal cells that constitutively express the Notch ligand Delta-like 1 (DL1) also confirmed that the raptor-deficient DN3 cells have impaired capacity to progress to the DP stage [105]. CD8-ISP cells have aberrant development into DP cells due to decreased proliferation. Raptor-deficient DN3 cells have impaired glycolysis, increased oxidative metabolism, and deregulated ROS production [105]. The raptor-deficient DN3a and DN4 cells also have diminished Myc expression. These metabolic defects likely contribute to the defective DN to DP transition. The increased induction of reactive oxygen species (ROS) could affect lineage fate decisions. Indeed, cell culture studies showed that increased levels of ROS favor γδ-T cell expansion, whereas ROS scavengers instead enhance ratios of αβ- vs. γδ-T cells. By single-cell RNA-Seq analysis, it was also shown that the absence of raptor leads to the upregulation of Notch signaling. While raptor deficiency lowers anabolic metabolism in post-selection thymocytes, these cells have robust αβ−TCR and ERK signaling. These findings underscore that the mTORC1 pathway, but not strong ERK signaling, is crucial for upregulating anabolic metabolism. The phosphorylation of S6 and 4E-BP1 is diminished in Rag-deficient DN3a cells, which have defective pre-TCR signaling due to an impaired TCRβ-chain rearrangement, indicating that mTORC1 signaling is coupled to the pre-TCR. Additionally, mTORC1 mediates Notch signals, since DN3a cells cultured with OP9-DL1 stromal cells that constitutively express DL1 have augmented phosphorylation of mTORC1 targets compared to those that were grown on feeder cells without DL1 [105]. Altogether, these findings support that mTORC1 plays a crucial role during the early stages of thymocyte ontogeny, particularly during the proliferative phases. During those stages, mTORC1 couples signals from the pre-TCR and Notch and promotes anabolism. Nonetheless, metabolic targets of mTORC1 signaling that could mediate its function during the different stages of development remain to be elucidated.

#### 5.1.2. The Role of mTORC2 in αβ-T Cell Development

The ablation of the mTORC2 components, rictor and SIN1 reveals that this complex is also involved in efficient thymocyte development. Using Cre controlled by the proximal promoter of *Lck* (*Lck-Cre*), rictor was abrogated starting at the DN2 stage and thereafter [119,120]. Deleting rictor reduces thymic cellularity due to impaired proliferation of DN subsets. However, mature T cells develop and transit to the periphery, suggesting that rictor is not absolutely essential for T-cell maturation. The ablation of rictor at DN2 leads to a partial block at the DN3 stage, thus diminishing every subset thereafter. Similar phenotypes are observed using the *Mx1-Cre* or *vav-Cre* systems, which both abrogate rictor at earlier stages of thymocyte development than does *Lck-Cre* [119,121]. In addition, there is a partial block at the CD8-ISP stage, resulting in a reduced proportion and number of DP thymocytes [120]. The partial block at the DN3 and CD8-ISP stages is accompanied by diminished expression of receptors required for thymocyte maturation, such as Notch1, pre-TCR and αβ−TCR, CD147, CD8 and CD4. There is increased post-translational misprocessing of TCRα in the absence of rictor, suggesting that the decreased surface levels of αβ−TCR could be due to improper protein folding. However, not all cell surface receptors are diminished in the absence of rictor. For example, CD127 (IL-7Rα) is elevated, suggesting that mTORC2 distinctly modulates the expression of specific cell surface proteins [120].

Deficiency in SIN1 also impairs early thymocyte development, but the phenotype was influenced by the mode of SIN1 ablation. In chimeric mice, wherein fetal liver cells from SIN1^−/−^ embryos were transplanted into WT congenic mice, there were no obvious defects in the development of thymic T-cell populations. The culture of SIN1^−/−^ fetal liver HSCs on OP9-DL1 stromal cells also revealed no defects in the development of DN1-DN4 subsets, as well as changes in TCR β-chain expression. While the differentiation of CD4^+^ helper cells is normal, there is increased proportion of Foxp3^+^ natural Treg (nTreg) cells in the thymus, indicating that mTORC2 negatively regulates the development of nTregs [122]. When SIN1 is ablated using the *Lck-Cre* promoter, a smaller thymus size, increased DN percentage, altered DN subsets and reduced total thymocyte number are observed [123]. Similar to rictor deficiency, abrogating SIN1 may affect the transition from the DN3 to DN4 stage. The reduced thymocyte numbers are due to decreased proliferation and not because of increased apoptosis. Glycolysis and OxPhos genes are downregulated in SIN1-deficient DN thymocytes, leading to decreased glycolytic capacity as compared to WT DN cells. The SIN1-dependent differentiation of DN thymocytes is mediated by the M2 isoform of pyruvate kinase (PKM2), one of the rate-limiting enzymes in glycolysis that catalyzes the transfer of phosphoenolpyruvate (PEP) to pyruvate. PKM2 expression is modulated via Akt, which promotes the nuclear translocation of PPAR-γ, a transcription factor that controls PKM2 expression. Pharmacological treatment to increase PKM2 activity or stimulate PPAR-γ rescues both the proliferation and development of SIN1-deficient DN thymocytes. Together with the findings on rictor-deficient thymocytes, the above studies demonstrate that mTORC2 is involved during the DN3/DN4 to DP transition. This transition, along with the CD8-ISP stage, is characterized by increased proliferation, suggesting that mTORC2 promotes metabolic reprogramming during these highly proliferative stages. The distinct regulation of metabolism by mTORC2 as compared to mTORC1 warrants further investigation.

#### 5.1.3. PI3K Signaling in αβ-T Cell Development

Thymocyte survival and proliferation during the β-selection checkpoint requires the integration of signals from different cell surface receptors such as the pre-TCR, Notch, IL-7Rα (CD127) and CXCR4. These receptors all activate the ubiquitous PI3K signaling, although the precise contribution of each receptor in triggering this pathway is unclear [124]. It is likely that proper thymocyte development requires optimal signal amplitude and duration provided by each of these receptors. For example, increased IL-7 signaling antagonizes Notch1 leading to decreased PI3K/Akt signaling and deregulation of thymocyte development, while B-cell lymphopoiesis is instead promoted in the thymus [125,126]. Pre-TCR signaling also antagonizes Notch1 expression after successful β-selection of DN3 cells [127]. Although little is known on how these receptors each contribute to the modulation of PI3K signaling, evidence is accumulating for the role of different PI3K isoforms and their downstream effectors in early T cell development.

PI3Ks are grouped into three major classes (I, II, and III) based on their structure and substrate specificity. Class I PI3Ks are further subdivided into Class 1A and 1B. The Class 1A are activated by receptor tyrosine kinases and include p110α, p110β, and p110δ. These catalytic subunits associate with regulatory subunits p50α, 55α, 85α, 85β, and 55γ. Class IB is activated by G protein-coupled receptors (GPCR) and consists of p110γ that associates with the regulatory subunits p101 or p84. The activation of PI3K results in the phosphorylation of phosphatidylinositol-4,5 bisphosphate (PIP2) to phosphatidylinositol-3,4,5-trisphosphate (PIP3). The increased levels of phosphorylated lipids attract signaling molecules to membrane moieties. Such signaling molecules, including PDK1 and Akt, have a PH domain that has affinity for the phosphorylated lipids. Their recruitment to this membrane compartment promotes their phosphorylation and activation. PDK1 phosphorylates Akt and other members of the AGC kinase family (such as S6K1) at their activation loop. This corresponds to Thr308 in Akt1, and the phosphorylation of this site activates this kinase. mTORC2 is also found in membrane compartments and its components SIN1 and possibly rictor have a PH domain [128,129]. PI3K activation enhances the mTORC2-mediated phosphorylation of Akt at its HM site, Ser473. This allosteric phosphorylation further enhances Akt activation. By modifying its substrate selectivity, maximal Akt triggering could yield broader target network and thus influence cell response and fate [130,131].

There have been conflicting reports on the role of PI3K in thymocyte development likely due to overlapping or compensatory roles of the PI3K isoforms. Early studies on ablation of the regulatory p85α revealed no obvious defects in T cell development [132,133]. Combined deletion of the four Class IA PI3K regulatory subunits, p85α, p55α, p50α and p85β in thymocytes also does not significantly affect T cell development [134]. In this combined deletion, the expression of the p110α and p110β catalytic subunits was consequently diminished. However, the loss of the regulatory p85α subunit alone was shown to impair the DN to DP transition in another system that focused on the analysis of the β-selection process [135]. Furthermore, the expression of a constitutively active form of p85α or a truncated version of p110α, that both result in PI3K activation, promotes positive selection of DP thymocytes without altering negative selection, suggesting that PI3K plays a role in the late stage of thymocyte maturation [136,137]. The two PI3K isoforms, p110δ and p110γ, play a more prominent role during early thymocyte development [138,139,140]. While the expression of a catalytically inactive form of p110δ does not affect T cell development in the thymus based on an unaltered CD4 vs. CD8 staining profile [141], the combined deletion of p110δ and p110γ results in a profound block at the β-selection checkpoint. This is accompanied by a block in DN3 expansion and increased apoptosis. The defects in the combined p110δ and p110γ-deficient mice are recapitulated by the combined deletion of p110δ and the class IB regulatory subunit p101 to which p110γ associates. These findings support that GPCR triggers PI3K during early αβ-thymocyte development. Indeed, stimulation of DN3 thymocytes with the chemokine CXCL12 that binds to the GPCR family member CXCR4 induces signaling that mainly involves the PI3K subunit p110γ. In contrast, signals from the pre-TCR are coupled to p110δ but not p110γ [124]. Together, these findings reveal that the presence of both Class IA and Class IB PI3Ks are required for optimal maturation of the αβ-lineage. The different cell surface receptors that activate distinct PI3K could allow for the spatial and temporal modulation of proliferation during thymocyte development.

The PI3K/Akt pathway also mediates Notch signaling. Notch signals are required to maintain PI3K-mediated glucose uptake and metabolism in DN3 cells [142]. Since Notch is a transcriptional regulator, its link to PI3K/Akt signaling is unclear. Notch1 could indirectly enhance PI3K/Akt signaling via the modulation of the transcription factor HES1, which represses PTEN expression [143]. In the absence of Notch signals, the expression of constitutively active Akt allele rescues glucose metabolism and overcomes the block in thymocyte development associated with the disruption of Notch [142].

Given the importance of PI3K signals in thymocyte development and proliferation, maintaining sufficient levels of signals from this pathway is critical to avoid malignancy and autoimmunity. PI3K signals are antagonized by PTEN, a tumor suppressor with lipid phosphatase activity. The loss of PTEN upregulates PI3K/Akt signaling and enables αβ-lineage thymocytes to bypass IL7 and pre-TCR-mediated signaling without compromising the levels of neither DN nor DP subsets [144]. However, while thymic cellularity is normal at birth, over time the loss of PTEN eventually leads to the development of lymphoma [144]. In addition to PTEN, PI3K/Akt signaling is also repressed by Itpkb (inositol-trisphosphate 3-kinase B) which produces IP4, a soluble antagonist of PIP3. Thus, Itpkb also modulates thymocyte development by reducing pre-TCR-induced PI3K/Akt signaling [145]. In contrast, Itpkb deficiency increases mTOR/Akt signaling and promotes thymocyte maturation to the DP stage even in the absence of Notch.

Increased PI3K signaling enhances PDK1-mediated phosphorylation of Akt and other AGC kinase family members. Hence, PDK1 also plays a crucial role in thymocyte development. The loss of PDK1 results in a block at the DN3 to DP transition [146]. PDK1 promotes the expression of key nutrient receptors such as CD71 (transferrin receptor) and CD98 (subunit of L-amino acid transporters) [147]. PDK1 mediates Notch signals and is essential for trophic and proliferative responses in thymocytes. A PDK1 mutant that can activate Akt but not the other AGC kinases rescues the expression of CD71 and CD98 as well as the differentiation to DP and SP cells. However, thymus cellularity remains low in this transgenic mouse model, indicating that other PDK1 targets could play a role in optimal thymocyte proliferation or survival.

Akt, a downstream target of PI3K signaling, plays crucial roles in thymocyte development. Akt has several isoforms, which possibly have redundant roles and a myriad of targets that are mainly involved in cell metabolism, survival and proliferation. The loss of Akt1 by itself does not have significant effects on thymocyte development likely due to compensation mechanisms by the other Akt isoforms [148]. Indeed, combined deletion of Akt1 and Akt2, and to a lesser extent Akt1 and Akt3, leads to more profound defects in thymocyte development beginning at the DN3 to DP transition [148,149]. The combined loss of Akt1 and Akt2 blocks proliferation of DN4 cells and diminishes DP survival, while the combined deletion of all Akt isoforms inhibits survival of all DN thymocytes [149]. For optimal activation, Akt is phosphorylated by mTORC2 at the allosteric Ser473 site. The phosphorylation of this residue upon TCR/CD3 stimulation is abolished in the absence of rictor [120]. On the other hand, the expression of Akt mutants revealed that the mTORC2-mediated phosphorylation of Akt is also important for Notch-induced thymocyte differentiation [119]. Notch activates Akt via an indirect mechanism involving the induction of HES1, which represses the transcription of PTEN. Akt has numerous metabolic targets as revealed in studies from cancer models and other tissues. In thymocytes, it mediates the increased expression of PFKFB3 and PFKFB4 during mitogen stimulation of thymocytes [150]. It also phosphorylates PFKFB3 at Ser461 in vitro although the role of this phosphorylation remains to be further examined. The inhibition of Akt decreases lactate levels, cell proliferation and protein synthesis concomitant with decreased levels of PFKFB3/4 and thus further supports the role of Akt in promoting glycolysis during thymocyte proliferation.

The ribosomal S6 kinase (RSK) is another AGC kinase that is modulated by PDK1. RSK has two kinase domains, one is homologous to the AGC kinases and another one to the CAMK family. The deletion of rictor in T cells decreases the phosphorylation of RSK at the HM site that is common in several AGC kinases [151]. Interestingly, mTOR kinase activity is not required for RSK regulation, instead mTORC2 could serve as a scaffold to modulate RSK activity or targets. Pharmacological inhibition of RSK using BI-D1870, which inhibits all four RSK isoforms does not prevent the DN to DP differentiation but blocks the Notch-induced proliferative expansion of these pre-T cells [147]. Genetic studies to ablate different RSK isoforms in thymocytes should provide better insights on their role in early T cell ontogeny.

#### 5.1.4. Other Metabolic Regulators of αβ-T Cell Development

The role of metabolism in modulating early T cell development is further highlighted by studies that ablate genes involved in metabolic regulation such as transcription factors, signaling molecules and metabolic enzymes. The function of the transcription factor c-Myc (also referred to as Myc) in metabolism is well characterized in peripheral T cells but we are only beginning to gain insights on its role during the metabolic reprogramming that accompanies T cell ontogeny. Highly proliferating DN thymocytes but not the quiescent DP cells are dependent on the expression of Myc [103,152]. Myc expression is elevated after β-selection but is downregulated at the DN4 stage and is largely undetectable by the DP stage [103,105]. Deletion of Myc at the DN1 stage also leads to similar defects as the ablation of raptor, resulting in decreased DP cells due to a block at the DN to DP transition [105]. Although no significant phenotype is observed upon deletion of Myc at the DP stage, it increases the efficiency of positive selection and the generation of CD8 SP thymocytes when overexpressed in DP cells of HY TCR transgenic mice [153]. Hence, downregulation of Myc could be a prerequisite for the DP stage. The expression of c-Myc is modulated by the transcriptional and epigenetic regulator bromodomain protein 4 (BRD4). BRD4 does not seem to be required for DN differentiation nor DP to SP maturation but instead is specifically essential for regulating cell cycle and metabolic pathways during CD8-ISP maturation [102]. BRD4 could thus be required for the transition from the highly proliferative DN to the quiescent DP stage. Why BRD4 is required specifically at the ISP stage whereas Myc is essential at both DN and ISP stages remains to be examined. The transcription of Myc is also controlled by an N-Me enhancer region whose deletion leads to a defective DN3 to DP transition as well as reduced number of DP and SP cells [154]. Myc expression is also reduced in DN3, DN4, and ISP cells in the absence of N-Me. Post-transcriptional upregulation of Myc that results in DN to DP proliferation and differentiation is also mediated by WNK1, a kinase that is involved in the regulation of ion homeostasis [155]. WNK1 couples pre-TCR signals via OXSR1 and STK39 kinases, and the SLC12A2 ion co-transporter that are essential for Myc regulation.

In mice with genomic deficiency of the peroxisome proliferator-activated receptor-δ (PPARδ), a transcription factor that senses fatty acid intermediates, there is decreased thymus cellularity resulting from impaired proliferation at the DN3-DN4 stages [156]. These defects are accompanied by diminished rate of extracellular acidification, mitochondrial reserve, and reduced expression of genes that are involved in glycolysis, TCA cycle, electron transport chain, and lipid biosynthesis. Hence, these findings imply that PPAR-δ is involved in metabolic reprogramming during the proliferative phases of thymocyte development following the β-selection checkpoint. In contrast, deletion of the sterol regulatory element-binding protein cleavage-activating protein (SCAP), a scaffold protein that modulates SREBP and that is necessary for lipogenic metabolism does not affect thymocyte populations although it is required for peripheral T cell proliferation [105]. The deletion of the glycolytic regulator HIF1α also does not have significant effects on thymocyte subsets [105]. Future studies to investigate in more detail the role of transcriptional regulators of genes involved in metabolism should provide better insights on how metabolic reprogramming shapes thymocyte development.

Other signaling molecules that regulate metabolism have also been implicated in the control of thymocyte development. The liver kinase B1 (LKB1), which responds to energy stress, is required for survival and differentiation of TCR-β selected DN thymocytes [157]. Knockout of LKB1 using the *Lck-Cre* system leads to a dramatic decrease in thymocyte numbers. DN cells accumulate at the DN3 stage and LKB1-deficient thymocytes have decreased numbers of DP and SP subsets [158,159]. LKB1 is involved in positive selection of DP thymocytes via its role in coupling signals from the αβ−TCR to downstream pathways [160,161]. Following TCR stimulation, LKB1 interacts with proximal TCR signaling molecules such as LAT and phospholipase C-γ1 (PLCγ1). Thus, LKB1 promotes the phosphorylation of PLC-γ1 and its recruitment to the LAT signalosome. LKB1-deficient thymocytes are more sensitive to glycolytic inhibitors and have decreased levels of Bcl-xL. Increasing the expression of Bcl-xL restores cellularity, the block at the DN3 stage and positive selection of SP cells. The expression of a constitutively active mutant of AMPK also restores DP cells from LKB1 deficiency-induced cell death [159]. LKB1 is also required for the expression of CD98 and for pre-TCR induced S6 phosphorylation [157]. When LKB1 is deleted at the DP stage, survival of those cells is not affected but the kinase seems to be necessary for DP differentiation [161]. The loss of AMPKα1 specifically at the DP stage does not affect thymocyte maturation and thus argues against a role for this kinase in mediating LKB1 functions during this developmental stage. Taken together, these findings reveal that LKB1 is required for thymocyte development and survival.

Changes in the expression levels of nutrient transporters during different stages of development further underscore the role of nutrient metabolism in thymocyte ontogeny. The glucose transporter GLUT1 is upregulated on metabolically active mouse and human thymocytes, while its deficiency leads to a significant loss in the total cell number [110,162,163]. Deficiency in amino acid transporters, such as Slc7a5 (aka LAT1), which transports large neutral amino acids including leucine, as part of a heterodimeric complex with CD98 (Slc3a2), does not seem to affect T-cell development in the thymus [164]. This is likely due to the presence of redundant transporters that compensates for the absence of Slc7a5. The mutation or targeted deletion of CD98 results in embryonic lethality [165]. CD98 is upregulated in DN cells but not in DP or SP thymocytes [166]. Since studies performed in non-T lineage cells reveal that CD98 has functions other than amino acid transport, such as the control of cell adhesion and proliferation, future studies should address its precise role during thymocyte development. The transferrin receptor CD71, which mediates iron entry, is required for thymocyte differentiation and is upregulated in cycling cells [163,167,168]. The phosphate transporter Pit2 is expressed in murine thymocytes with high metabolic activity such as DN3b, DN4 and CD8^+^-ISP cells [109]. In human thymocytes, Pit2 was increased during early post-β selection CD4^+^-ISP and αβ−TCR^+^CD4^hi^DP thymocytes that co-express CD71, thus suggesting it is also present during high metabolic stages. Interestingly, another phosphate transporter, Pit1, is found in mature CD3^+^ murine and human thymocytes and is a biomarker of an aging thymus [109]. In the mouse, these cells are mostly γδ−TCR and NKT thymocytes by 1 year of age (equivalent to 50–60 years as a human). These findings not only support the selective expression of nutrient transporters at different stages of development but also suggest that metabolic changes occur during thymocyte aging.

Deletion of genes whose products are directly involved in metabolic processes also provide support on the crucial role of metabolism in thymocyte development. Deficiency in the mitochondrial pyruvate carrier (MPC1), which mediates uptake of pyruvate into mitochondria, leads to impaired DN β-selection, DN to DP transition and positive selection of DP cells [169]. mTORC1, myc, and Akt signaling are also affected under these conditions. Ablation of the optic atrophy 1 (OPA1) gene, a mitochondria-shaping protein that functions in the fusion of inner mitochondrial membrane (IMM) and mitochondrial respiration impairs DN3 thymocyte maturation [106]. The OPA1-deficient T cells that reach the periphery are also metabolically and functionally defective. Deficiency in the ribosomal protein L22 (rpl22), an RNA binding protein and component of the 60S ribosomal subunit, blocks differentiation of αβ-lineage precursors past the β-selection phase [170]. Remaining rpl22-deficient cells that passed the β-selection checkpoint have increased apoptosis. The loss of rpl22 is associated with increased p53 biosynthesis. The combined loss of rpl22 and p53 alleviates development defects that are associated with rpl22 deficiency alone. The loss of rpl22 induces the expression of genes involved in apoptosis, some of which are p53 effectors such as PUMA [171]. Whether rpl22-deficient thymocytes have metabolic defects remains to be investigated. The role on thymocyte development of other transporters of intracellular metabolites and key metabolic enzymes/effectors on thymocyte development awaits further investigation.

Metabolism also generates products that are utilized for co- or post-translational modifications of proteins. An example of this is the metabolite UDP-GlcNAc, which is a product of the hexosamine biosynthetic pathway. Glucose and glutamine are utilized to produce UDP-GlcNAc, which is used for intracellular protein *O*-GlcNAcylation and surface/secretory protein *N*-glycosylation. Protein *O*-GlcNAcylation is upregulated at the DN3-DN4 transition whereas DP cells have lower levels of *O*-GlcNAcylation. Notch stimulation enhances glucose and glutamine uptake and increased intracellular protein *O*-GlcNAcylation. Ablation of *O*-GlcNAc transferase (OGT) results in reduced thymocyte numbers due to decreased DP cells as well as positively selected CD4-SP or CD8-SP thymocytes [107]. In vitro culture experiments rescue the differentiation defects, but the proliferation remains impaired, despite the presence of IL-7 and Notch signaling. Using the *vav-Cre* system to abrogate *O*-GlcNAcase (OGA), the enzyme that removes *O*-GlcNAc, also leads to a dramatic decrease in cell numbers at all stages of thymocyte differentiation [172]. Deficiency of fucosyltransferase (Fut8), which is involved in *N*-glycosylation of cell surface receptors, leads to reduced phosphorylation of the tyrosine kinase ZAP70 in DP cells due to the loss of TCR fucosylation [173]. The attenuated downstream signaling in absence of Fut8 reduces DP and SP cell numbers and thus decreases the size of the Fut8^-/-^ thymus. Together, these findings provide evidence that thymocyte development is also dependent on metabolism because of metabolites, such as UDP-GlcNAc, that modulate critical regulatory proteins.

#### 5.1.5. Role of Autophagy

Autophagy, which includes macroautophagy, microautophagy, and chaperone-mediated autophagy, is a process that degrades proteins, macromolecules, and even organelles, thus allowing recycling of intracellular nutrients. Autophagy goes through five stages: induction, nucleation, elongation, lysosomal fusion, and degradation. These stages are regulated by protein complexes and negatively controlled by mTORC1 signaling. Autophagy plays a role in early T cell development in the thymus. This process is important during the selection of DP thymocytes as it is involved in peptide presentation by stromal cells and antigen presenting cells (reviewed in [174]). T lineage-targeted ablation of autophagy genes also indicates a role for this process in optimal T cell ontogeny, although its involvement at specific stages of thymocyte maturation remains to be further elucidated.

Fetal liver cells from Atg5-deficient mice transferred into lethally irradiated congenic hosts results in reduced numbers of reconstituted thymocytes but does not alter the percentage of the different subsets [175]. Conditional deletion of ATG3 using *Lck-Cre* also reduces total thymocyte numbers whereas the CD4/CD8 subsets had normal proportions [176]. Using the same promoter, deficiency in Atg7 also diminishes thymocyte numbers including SP cells although percentages are normal [177,178]. These findings indicate that autophagy is critical for the survival or proliferation of DN thymocytes as well as for the transition from the DN to DP stage. Furthermore, there is defective clearance of excess mitochondria during thymocyte egress to the periphery, resulting in elevated ROS production and Bcl-2 expression in peripheral naïve T cells [175,176]. These findings underscore the role for autophay in maintaining organelle homeostasis. When Atg genes are ablated at later stages of thymic development using *CD4-Cre* instead of *Lck-Cre*, there is no effect on thymocyte numbers [179,180].

Beclin1, is a key component of a PI3KC3 lipid kinase complex that is involved in autophagosome nucleation. In Beclin1-deficient Rag1^-/-^ chimeras, thymocyte numbers are significantly reduced although peripheral T cells are normal [181]. The decreased cellularity is likely due to a defect in the maintenance of thymocyte progenitors, indicating a role for Beclin1 during the early stages of T cell development. The class III PI3 kinase, Vps34 forms a complex with Beclin1 and thus plays a role in the autophagic induction stage as well as in endomembrane trafficking. Ablation of Vps34 using *Lck-Cre* reduces thymocyte numbers due to decreased cell survival [2], but when knocked out at later stages of thymocyte maturation using *CD4-Cre*, there is no change in total or relative thymocyte numbers [179,180]. The deficiency in Vps34 also does not impair mitochondrial clearance, suggesting that other regulators could compensate for its function or that it is not crucial for autophagy in the later stages of thymocyte maturation [2]. Altogether, these findings support a role for autophagy in thymocyte development by promoting survival, but more studies are needed to define its role in metabolic remodeling during the different stages of T cell ontogeny.

#### 5.1.6. Effects of Nutrients and Metabolites

The effect of diet and nutrient/metabolite supplementation on thymocyte populations highlights the role of nutrient metabolism in thymocyte development. Glucose is critical in triggering glycolytic enzymes during mitogen-induced thymocyte proliferation [182]. Whereas resting thymocytes undergo oxidative metabolism, mitogen-stimulated thymocytes utilize glucose and glutamine and produce lactate from glucose [183]. Glutamine does not seem to be utilized for lactate formation, but it is utilized for glutaminolysis and transaminase reactions in mitogen-stimulated thymocytes [184]. Mice fed with high fat diet (HFD) from weaning to early adulthood leading to obesity have decreased CD4^+^ and CD8^+^ SP cells [185]. The HFD mice have increased thymic triacylglycerol content and their thymic tissue has increased apoptosis and defective Akt/mTOR signaling. Administration of the antioxidant *N*-acetyl-*L*-cysteine (NAC) to mice prior to exercise prevents exercise-induced thymocyte apoptosis, alterations in glutathione levels and mitochondrial membrane depolarization [186], thus highlighting how oxidative stress negatively impacts thymocyte survival. More recent studies have unraveled the importance of maintaining redox homeostasis for proper αβ- and γδ-T cell development as well as highlighted the role played by mTORC1 in the regulation of this homeostasis [105].

Compartmentalization or cell crowding in the thymus also affect stimulatory signals as well as nutrient availability to the developing thymocytes. The importance of the thymic microenvironment is highlighted by the findings that altering mTORC1 signals in the TECs, which stimulate T cell ontogeny, leads to aberrant early thymocyte development. Indeed, using the *Foxn1-Cre* system to delete raptor in TECS diminished their numbers in both fetal and adult mice, as well as perturbed their maturation and affected the normal ratio of medullary TECs (mTECs) versus cortical TECs (cTECs) [118]. Thymocyte maturation and cellularity were severely impacted by this thymic atrophy. However, it remains unclear precisely by which molecular mechanisms mTORC1 controls TEC development and function and how alterations in these processes affect early thymocyte development.

### 5.2. γδ-T Cells

γδ-T cells develop in the thymus alongside the αβ-T cell lineage (Figure 3). However, some marked differences distinguish γδ- and αβ-T cell ontogeny. In stark contrast to αβ-T cells that egress the thymus as naïve T cells, γδ-T cells are functional effector cells by the time they leave the thymus. γδ-T cells emerge from DN2/3 thymocytes in the presence of high IL-7R signaling and develop into distinct effector lineages, including IFNγ-producing γδ-T1 and IL17-producing γδ-T17 cells [187,188]. In addition to γδ-T17 cells that primarily develop in the fetal thymus, distinct γδ-T cells exclusively emerge from that organ at the fetal age. They express TCRγ-chains specifically harboring the recombined variable segments, Vγ5 and Vγ6 in association with the TCRδ-chain, Vδ1. Thus, instead of an uninterrupted generation of cells bearing polymorphism for antigen recognition, γδ-T subsets arise in “waves” defined by the variable domain of their TCRγ-chain (Vγ), which further characterizes their functionality and tissue homing. How those waves are generated as well as how the development of γδ- and αβ-thymocyte diverge remains a subject of further investigation. So far, the strength of pre-TCR (αβ) versus γδ-TCR signaling is thought to determine lineage fate [189,190]. This model posits that relatively weaker signals promote αβ-T lineage whereas strong TCR signaling favor γδ-T cell development. Another model proposes a more stochastic mechanism wherein other signals determine cell fate prior to TCR expression. It is noteworthy that γδ-selection triggers less cell proliferation compared to β-selection [191]. γδ-selected DN3 thymocytes also have distinct gene expression changes and Notch/delta signaling is not essential for their development.

Ablating raptor by using the *CD2-Cre* system does not significantly affect the development of γδ-T cells [105]. In contrary, raptor-deficient mice have more γδ- cells than WT animals, indicating that mTORC1 suppresses γδ-T cell development (Figure 5). Raptor-deficient DN3 cells have higher phosphorylation of ERK as well as enhanced expression of EGR1 and ID3, supporting the notion that strong signals coupled by ERK promote γδ-T cell development. ERK modulates a distinct set of substrates via its DEF-binding pocket (DBP) to specify γδ-thymocyte differentiation while suppressing αβ-lineage development [192]. There is also higher percentage and cellularity of the CD73^+^ subset in raptor-deficient γδ-T cells as well as increased CD73 expression. Similarly, Myc-deficiency skews thymocyte development towards γδ-T cells as compared to the αβ-lineage due in part to increased ERK signaling. The deletion of Myc also increases the frequency and cellularity of the CD73^+^ subset and higher levels of CD73 expression is also observed [105]. Consistent with this, raptor deficient cells have reduced Myc levels, suggesting that mTORC1 modulates Myc expression and γδ-T cell ontogeny. Conversely, Myc-deficient cells have reduced phosphorylation of both downstream mTORC1 targets, S6 and 4E-BP1. Hence, these studies unravel that mTORC1 and Myc are dispensable for γδ-T cell development.

Emerging studies suggest that the metabolic requirements of αβ- vs. γδ-T cells are distinct during their development. Deletion of the mitochondrial pyruvate carrier 1 (MPC1) which compromises αβ-T cell ontogeny, does not affect γδ-T cell development but the latter “protective” mechanism remains to be examined [169]. Similarly, the loss of rpl22 does not disturb γδ-T ontogeny, while it affected tremendously αβ-lineage development [170]. Furthermore, blocking iron uptake with CD71 monoclonal antibody abrogates αβ-T cell development in fetal thymic organ cultures, whereas development of the γδ-lineage is unaffected. In addition, Vitamin B1 (Vit B1) deficiency favors the maturation of γδ-thymocytes whereas differentiation into αβ-DP cells is decreased [193]. Notably, the lack of Vit B1 leads to increased production of TGF-β superfamily members due to increased branched-chain α-keto acids in thymic stromal cells. Overall, the precise metabolic needs of developing γδ-T cells are less well understood as compared to αβ-T cell ontogeny and thus merits further investigation.

The role of metabolism in γδ-T cell development could in part be highlighted by manipulating their microenvironment. Thus, the specific abrogation of raptor in fetal TECs deregulates the temporal control of TCRVγ/δ recombination that normally generates IL17-producing γδ−T cells at this embryonic stage [118]. Instead, the raptor compromised-TECs signal for increased γδ−T17 differentiation only in adulthood. The metabolic changes that occur in TECs and how this affects nutrient availability or stimulatory conditions during γδ-T cell differentiation would need further examination.

Studies on epigenetic regulation could also shed light on how metabolism controls the divergence of γδ- vs. αβ-T cell development in the thymus. A transient increase in tri-methylation of Histone H3 at Lysine 27 (H3K27me3) without open chromatin modification occurs in the αβ-lineage during β-selection [97]. In contrast, owing to strong TCR signaling, emerging γδ-T cells display large changes in chromatin accessibility. Since chromatin modifications are dependent on metabolites, it would be crucial to dissect the metabolic processes that impact epigenetic regulation of αβ- vs. γδ-T cell differentiation.

### 5.3. Thymic Regulatory T Cells (tTregs)

Regulatory T cells (Tregs) play a crucial role in preventing autoimmunity since they prevent the expansion of self-reactive T cells. The natural CD25^+^Foxp3^+^CD4^+^ regulatory T cells have two subgroups; the thymic Tregs (tTreg, aka nTreg), which mature in the thymus and the peripheral Tregs, which differentiate in the periphery [194,195]. tTregs are generated in the thymus as a functionally mature T cell subset that have suppressive function even before antigen recognition (in contrast to induced Tregs, iTregs, that differentiate from naïve CD4^+^ T cells after antigen stimulation in the periphery). They represent 5–10% of circulating CD4^+^ T cells. Although most thymocytes that express high affinity TCRs for self-antigens are eliminated via negative selection in the thymus, those that express TCRs with intermediate/high affinity can differentiate into tTregs. The depletion of CD25^+^Foxp3^+^CD4^+^ Tregs leads to autoimmunity while reconstituting this population prevents autoimmune disorders [196]. The expression of the transcription factor Foxp3 is a key hallmark of Tregs, which controls their development and function. Deficiency in Foxp3 blocks the development of natural Tregs and enhances the activity of T cells with self-antigens and thus promotes lymphoproliferative disorders and multi-organ autoimmunity [196]. Epigenetic regulation of Foxp3 expression occurs in the thymus and its expression is a distinct marker of stable Tregs in the periphery [197]. Foxp3 induction is dependent on the repression of PI3K signaling, while it is promoted by rapamycin treatment [198]. Added in vitro, rapamycin expands naturally occurring murine CD4^+^CD25^+^Foxp3^+^ Tregs [199]. In vivo, it also increases the proportion of nTregs in the thymus and the periphery, and maintains their functionality in contrast to other immunosuppressive drugs such as FK506 and cyclosporin A (CsA), which both disrupt Treg function [200]. Inhibition of the PI3K/mTOR signaling network increases de novo expression of Foxp3, as well as Treg-like mRNA and miRNA profiles in thymocytes and peripheral T cells of the CD4 lineage [201]. Expressing the kinase inactive PI3K isoform, p110δD910A in mice leads to a 2-fold increase in the proportion of Foxp3^+^ cells in the thymus [202]. However, while this mutant enhances Treg development in the thymus, fewer mature cells are found in the periphery. p110δ regulates the expression of CD38, a transmembrane cyclic ADP ribose hydrolase, and the increased levels of this enzyme are linked to high Treg suppressive activity. PTEN expression is also maintained in tTreg to prevent high PI3K activity [203,204]. Altogether these findings unravel that dampening PI3K/mTORC1 signaling promotes thymic Treg differentiation.

mTORC2 signaling also curbs nTreg development. Rictor deficiency, but not PTEN deficiency in mice leads to a significant increase in thymic nTreg cells, supporting the role of mTORC2 in repressing the expansion of these cells [203,205]. In addition, the expression of activated Akt in the thymus during normal Treg differentiation prevents the generation of CD4^+^Foxp3^+^ cells without affecting the positive selection of CD4^+^ thymocytes [206]. Akt signaling is curtailed by the PH-domain leucine-rich repeat protein phosphatase (PHLPP). In Tregs, the expression of this phosphatase is upregulated [207]. In PHLPP^−/−^ mice, tTregs develop normally but have diminished suppressive capacity. However, another study found that OX40L & IL-2 trigger an mTOR/Akt-dependent proliferation of tTreg cells that occurs in the thymus during the TCR-independent phase of human and murine tTreg ontogeny [208]. Hence, stimulatory signals that affect mTORC2 signaling could influence its role in modulating tTreg generation, proliferation and suppressive function.

A more complex involvement of mTOR signaling during nTreg differentiation is further demonstrated by studies using a knockout of TSC, a negative regulator of mTORC1. Knockout of TSC1 using the *Lck-Cre* system also results in an increased percentage and cell number of thymic CD4^+^CD25^+^Foxp3^+^ nTregs [205]. The TSC1-deficient nTreg cells have an increased proliferative response to IL-2 via the transcription factor, STAT5. Rapamycin treatment further increases the frequency of nTreg in TSC1^−/−^, as well as rictor^−/−^ mice. These findings suggest that striking a balance in mTOR signaling, rather than simply diminishing its activity, is a prerequisite for proper nTreg homeostasis.

Unlike induced Tregs that prefer lipid oxidation to fuel mitochondrial OxPhos, tTregs favor glycolysis and glutaminolysis despite maintained Foxp3 expression [204]. In the presence of TGF-β, tTregs diminish PI3K-mediated mTOR signaling, as well as glucose transporter and Hexokinase 2 (Hk2) expression, resulting in the reprogramming of metabolism towards OxPhos. Glut1 deficiency does not affect CD4^+^Foxp3^+^CD25^+^ nTreg cellularity but it reduces Foxp3-negative CD4^+^ cells. This reduction in the latter progressively increases the amounts of Foxp3^+^ nTreg reaching the peripheral T cell compartment over time [110]. When PI3K activity is increased in tTregs due to the loss of PTEN, aerobic glycolysis rises during the early phase of cell activation, but it is later restored to WT levels. This is accompanied by increased metabolites generated from nucleotide biosynthesis and lipogenesis. In contrast, iTregs lacking PTEN upregulate aerobic glycolysis while maintaining OxPhos, which is coincident with reduced Foxp3 expression. These findings highlight the metabolic differences between tTregs and iTregs.

## 6. Perspectives and Conclusions

The mTOR complexes have broad functions in the control of cellular metabolism. Studies on their role in early T cell development have revealed that, while the disruption of their components, raptor (mTORC1) or rictor and SIN1 (mTORC2) may not be absolutely required for the generation of naïve αβ-T cells, they are necessary for optimal thymocyte development and expansion. These complexes are critical for the highly proliferative stages of αβ-lineage development, likely due to their role in the control of nutrient uptake and metabolism. These two essential biological processes are upregulated in cell proliferation in order to provide energy and building blocks necessary for growth and expansion. Furthermore, although inactivation of mTOR favors Treg development [209], more recent studies reveal that this kinase is not totally dispensable for their development in mice and that it functions in maintaining the suppressive function of Tregs [94,210]. The role of the mTORCs and metabolism in the development of distinct γδ-T-cell subsets would also need to be further examined. While the strength of signals from the TCR and other cell surface molecules influence cellular fate [211], how these signals reprogram metabolism as well as how the cellular microenvironment and nutrient availability also contribute to lineage choices in T cell ontogeny need further investigation. The finding that a subset of proinflammatory T cells co-express both αβ- and γδ-TCR [212] may suggest that the metabolic microenvironment may have a more prominent role in controlling T cell fate. The metabolites produced by cellular metabolic processes also impact regulation of the genome and thus affect gene recombination and expression. Epigenetic regulation of V(D)J recombination via histone methyltransferases suggests that metabolic intermediates that are utilized for methylation are critical for thymocyte development [213]. Thus, the contribution of metabolism and signaling pathways such as mTOR that modulate metabolism on early T cell development deserves further investigation.

Understanding the role of mTOR complexes and metabolic processes in early T cell development has implications for preventing allograft rejection, improving therapeutic strategies for T cell lymphoma, and immunotherapy for cancers and autoimmune disorders [214]. Rapamycin is used in the clinic as an immunosuppressant to prevent allograft rejection. Clinical trials are underway for the use of rapamycin analogs in autoimmune disorders. Results from these trials and other human studies have already revealed promising results [215,216,217,218,219]. Antimetabolites that reprogram cellular metabolism and thus mitigate mTORC1 signaling are also being examined for the treatment of autoimmune disorders. *N*-acetylcysteine, which mitigates ROS and downregulates mTORC1 signaling in T cells, shows promise for systemic lupus erythematosus [220]. As we expand our knowledge on specific metabolic requirements during T cell ontogeny as well as the metabolic needs of different T cell subsets, we can develop more effective therapeutic strategies using such anti-metabolites or small molecules that target metabolic processes.

The generation of T cell lymphoma also often occurs during the highly proliferative DN stage of thymocyte maturation, emphasizing the importance of proper regulation of metabolism during this developmental phase. The inhibition or disruption of mTOR complexes prevents the development of lymphoma. Thus, rictor deficiency significantly, but not completely, suppresses Notch-driven T-ALL [119]. On the other hand, ablation of raptor in the T-ALL model induces cell cycle arrest and eradicates leukemia [114]. The disruption of genes involved in mTOR signaling and the control of key metabolic pathways also prevents the generation of lymphoma that is triggered by oncogenic mutations. Future studies should unravel how specific oncogenic mutations could remodel metabolism and how this eventually leads to malignancy.

The importance of metabolism in early T cell development underscores the contribution of nutrition in this process. The effects of diet on the immune system and T cells in particular have been well documented [221]. Nutrient effects on age-related thymus atrophy and thymocyte ontogeny have also been reported [222]. Aging not only impacts the generation of T cells but also attenuates immune responses in general. Thymic involution, characterized by decreased size, cellularity and functional defects occurs during the aging process. The decline in the immune system is associated with increased susceptibility to infections, inefficient response to vaccination and prevalence of cancers [223]. Metabolic changes that occur during aging likely influence T cell development in the thymus. For example, expression of the phosphate transporter PiT1, but not PiT2, was found on CD3^+^ thymocytes from aged mice, suggesting metabolic changes and adaptations occur during thymic aging [109]. It is also interesting to note that while defects in negative selection of DP thymocytes increase with age, the generation of tTreg is not diminished but actually enhanced [224]. Rapamycin has been shown in various animal models to prolong lifespan. Whether it could have benefits in preventing age-related thymic involution would need to be further explored. In addition, further investigations will be necessary to understand the metabolic processes as well as the signaling pathways that control metabolism of the thymus microenvironment.

## Figures and Tables

**Figure 1 genes-12-00728-f001:**
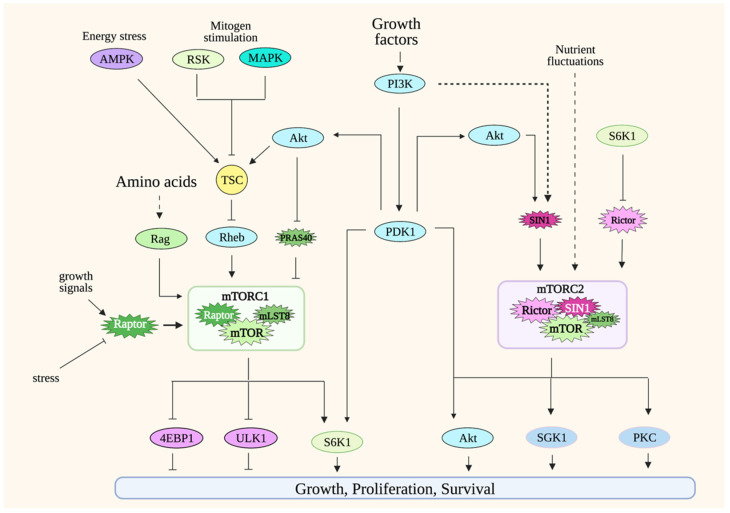
Simplified scheme of mTORC1 and mTORC2 signaling. mTORC1 is active in the presence of nutrients such as amino acids. Its activation is potentiated by growth factors/PI3K signaling. Modulation of mTORC1 occurs at many levels, including via post-translational regulation of its components such as raptor and via the GTPases Rag heterodimers and Rheb. Other environmental signals are conveyed to mTORC1 via the tuberous sclerosis complex (TSC), which acts as a break for mTORC1 signaling during unfavorable growth conditions. mTORC2 is activated by growth factors/PI3K signaling and during nutrient fluctuations. Less is known on mTORC2 regulation, but so far, its components rictor and SIN1 undergo posttranslational modifications that modulate mTORC2 activity. mTORC1 phosphorylates numerous substrates to either promote anabolic metabolism or inhibit catabolic processes. mTORC2 phosphorylates substrates that are involved in proliferation or promoting cell survival during stress conditions. Solid lines indicate direct regulation; dashed lines indicate more indirect regulation. Refer to Abbreviations.

**Figure 2 genes-12-00728-f002:**
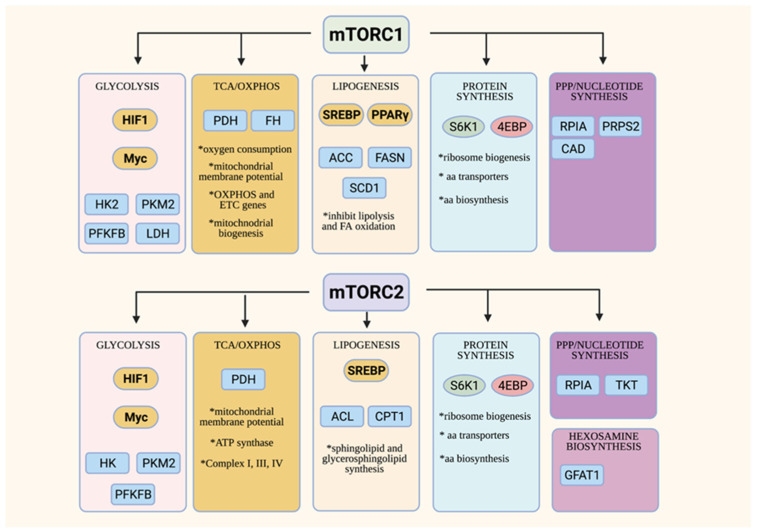
Overview of metabolic functions of mTORC1 and mTORC2. Both the mTORC1 and mTORC2 control metabolic processes via modulating transcription factors that are involved in metabolic pathways (in yellow ovals) and/or the control of other metabolic enzymes or transporters (in blue squares).

**Figure 3 genes-12-00728-f003:**
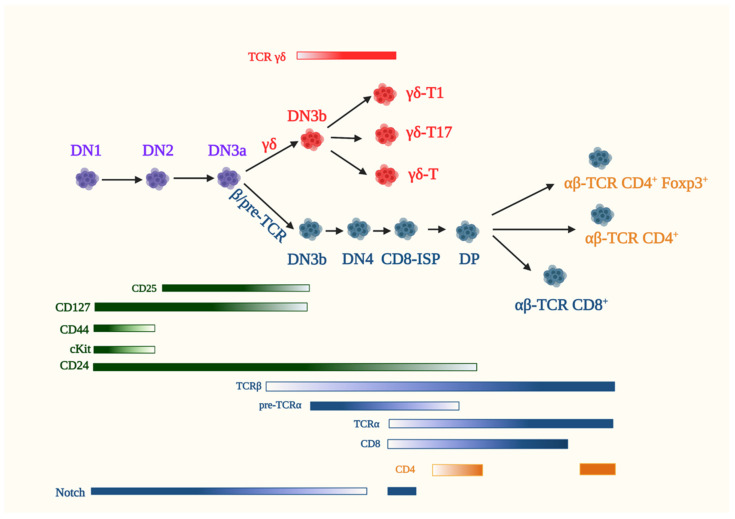
T-cell development in the thymus. Two major types of T cells develop in the thymus, the αβ- and γδ-T cells.

**Figure 4 genes-12-00728-f004:**
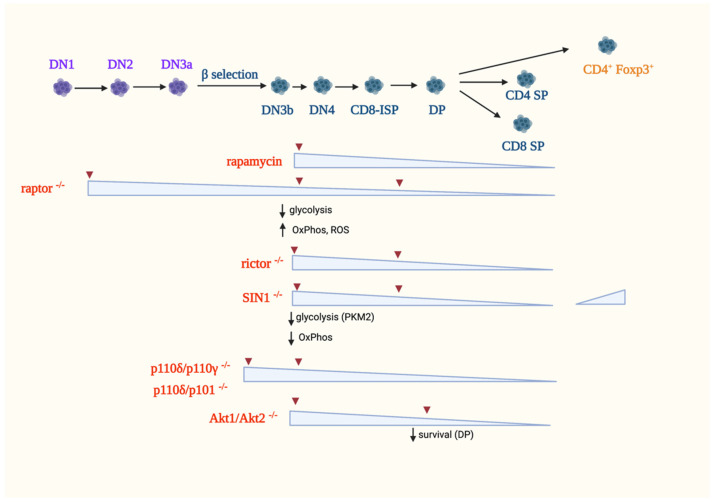
Effect of mTOR signaling disruption on metabolism and early αβ-T-cell development. Gene disruption or rapamycin treatment decrease the thymocyte numbers (encompassing stages indicated by blue triangles; downward slope indicates a decrease, while upward slope represents an increase) due to impaired progression at specific stages (stage of impairment indicated by small, brown triangles). In contrast, SIN1 disruption increases the tTreg (CD4^+^ Foxp3^+^) numbers.

**Figure 5 genes-12-00728-f005:**
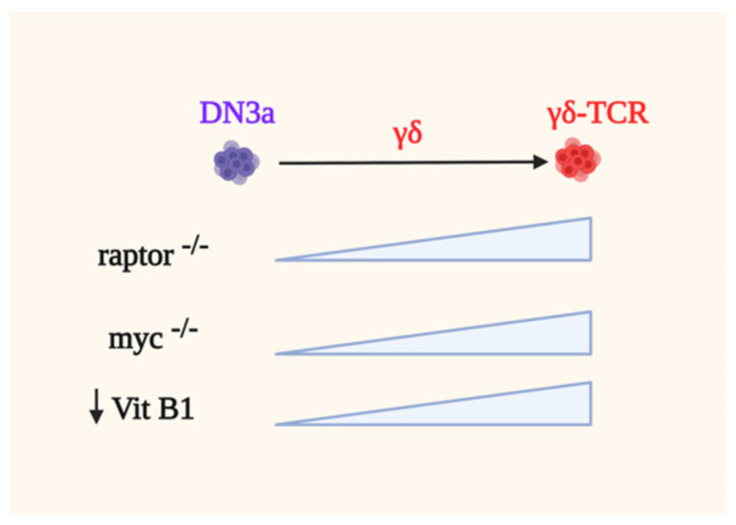
γδ-T cell development does not have the same metabolic requirements as αβ-T cells (see Figure 4), although their expansion can be enhanced during impaired αβ-thymocyte development.

## Data Availability

Not applicable.

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
