# Peer review of "MTOR Signaling and Metabolism in Early T Cell Development"

_genes, 2021, doi:10.3390/genes12050728_

Round 1
Reviewer 1 Report
This is a very well written review article. The authors have presented all the relevant information in a well synthesized manner. Only a few minor comments:
A list of abbreviations would be very useful.
Some of the figures are not cited in the text.
In Figure 1, what do the solid and broken lines indicate?
Line 354: discussed in Chapter 5.2?
Figure 4: No blue and brown triangles are visible in the Figure.
Line 646: How something can decrease the absence of another thing?
Figure 5: No comparison is made between αβ and γδ T cell development requirements.
Author Response
(Our responses are Italicized)
This is a very well written review article. The authors have presented all the relevant information in a well synthesized manner. Only a few minor comments:
We thank the reviewer for the positive review and appreciate all the comments and suggestions. We also thank the reviewer for his time.
A list of abbreviations would be very useful.
We have added this now as Appendix A.
Some of the figures are not cited in the text.
We have corrected this now.
In Figure 1, what do the solid and broken lines indicate?
We indicated this now at the end of Figure Legend 1 (Lines 164-165)
Line 354: discussed in Chapter 5.2?
We have corrected this now.
Figure 4: No blue and brown triangles are visible in the Figure.
We have clarified this now in Figure Legend 4 (lines 467-469)
Line 646: How something can decrease the absence of another thing?
We have changed the wording to “is abolished”.
Figure 5: No comparison is made between αβ and γδ T cell development requirements.
We have clarified Figure legends 4 and 5. Figure 4 indicates developmental effects of disruption of mTOR signaling during ab-T cell development whereas Figure 5 shows emerging data indicating the effects of disrupting mTOR signaling on gd-T cells.
Reviewer 2 Report
The work concerns the role of mTOR complexes in the differentiation of T cells. This issue is very interesting and timely, however, the work requires corrections.
Major comments:
1) The manuscript is extremely long. It describes the complex molecular mechanisms linking mTOR activity with T-cell activity and differentiation, so it is obvious that such descriptions require a large amount of space. However, I have the impression that in some chapters the authors departed from the essence of the work, devoting a lot of space to descriptions that have already been deeply discussed in the literature. A typical example is the descriptions of intracellular signaling relating to mTOR. Regulation of this kinase activity by various pathways has already been described in detail in the literature (the PubMed database shows over 25,000 records). Examples of such review articles are: (1) Saxton & Sabatini (2017) Cell 168(6):960-976; (2) Zou et al (2020) Cell Biosci 10:31; (3) Wang & Zhang (2019) Adv Exp Med Biol 1206:67-83; (4) Pierzynowska et al (2018) Metab Brain Dis. 33(4):989–1008. A similar note applies to the chapter on the function of mTOR in cell metabolism (also about 25,000 records in databases).
My thoughts on the descriptions of the precise regulation of mTOR activity (lines 108-185 and 198-237) as well as the role of mTOR in cellular metabolism (lines 238-325) could be replaced by short paragraphs which would only list basic information about the main pathways regulating mTOR activity and its role in metabolism with appropriate citations. Possibly, a broader description of these pathways should be left, which are important in terms of the influence of mTOR on T cells. It is important because currently the work is not very focused and departs from the main topic.
2) In the description of mTORC1 activity in the chapter “mTOR signaling”, it is mentioned that the main role of mTOR is to control protein biosynthesis. This is true, but it should be clarified that the main role of mTORC1 is the switch between autophagy and protein biosynthesis. However, only one sentence is devoted to autophagy in this description. I believe that this description should be expanded and, moreover, more attention should be paid to it, as it is known that autophagy itself also regulates the development, differentiation and activation of T cells. See for example: (1) Bronietzki et al (2015) Immunol Cell Biol 93(1):25-34; (2) Merkley et al (2018) Front Immunol 9:2914.
3) The abstract is incorrectly written. Too much space has been devoted to the description of the selection of T lymphocytes and their formation, and too little to the role of mTOR in regulating the differentiation of T lymphocytes and thymocytes. The description of the formation of the immune system cells themselves (lines 12-21) should be shortened and the description of a more precise role of mTOR and its regulation in the described processes of the selection of these cells extended, which is the main subject of this manuscript (lines 22-25). The abstract should be a summary of the article and not an introduction to the main topic.
Minor comments:
4) The authors at the end of some chapters provide a plan of descriptions in the following chapters (lines 47-55; 323-325; 431-437). I don't think such plans are needed. This distracts the reader from the continuity of the text and generates unnecessarily extra space in the already long manuscript.
5) Please check the spacing. Throughout a manuscript, it seems to me that there is often a double space before starting a new sentence.
6) There is a lack of complete affiliation data.
Author Response
(Our reply is Italicized)
The work concerns the role of mTOR complexes in the differentiation of T cells. This issue is very interesting and timely, however, the work requires corrections.
We appreciate the positive comments and the careful review of our manuscript from Reviewer 2. We have addressed all comments and suggestions. We also thank the reviewer for his/her time.
Major comments:
1) The manuscript is extremely long. It describes the complex molecular mechanisms linking mTOR activity with T-cell activity and differentiation, so it is obvious that such descriptions require a large amount of space. However, I have the impression that in some chapters the authors departed from the essence of the work, devoting a lot of space to descriptions that have already been deeply discussed in the literature. A typical example is the descriptions of intracellular signaling relating to mTOR. Regulation of this kinase activity by various pathways has already been described in detail in the literature (the PubMed database shows over 25,000 records). Examples of such review articles are: (1) Saxton & Sabatini (2017) Cell 168(6):960-976; (2) Zou et al (2020) Cell Biosci 10:31; (3) Wang & Zhang (2019) Adv Exp Med Biol 1206:67-83; (4) Pierzynowska et al (2018) Metab Brain Dis. 33(4):989–1008. A similar note applies to the chapter on the function of mTOR in cell metabolism (also about 25,000 records in databases).
My thoughts on the descriptions of the precise regulation of mTOR activity (lines 108-185 and 198-237) as well as the role of mTOR in cellular metabolism (lines 238-325) could be replaced by short paragraphs which would only list basic information about the main pathways regulating mTOR activity and its role in metabolism with appropriate citations. Possibly, a broader description of these pathways should be left, which are important in terms of the influence of mTOR on T cells. It is important because currently the work is not very focused and departs from the main topic.
We have now shortened the section on mTOR signaling (Section 3) but retained some information that are pertinent to understanding the latter discussion on the role of mTOR in early T-cell development. We also tried to trim the discussion on mTOR functions in metabolism (section 4) but we find it important to give some examples of how the mTORCs control different aspects on metabolic processes. There are indeed numerous reviews on metabolism but there is really not much that discuss metabolism in the context of both mTOR complexes. Since we are only providing an overview, we have now added references to recent review articles that discuss in more detail the role of the two mTOR complexes in metabolism.
2) In the description of mTORC1 activity in the chapter “mTOR signaling”, it is mentioned that the main role of mTOR is to control protein biosynthesis. This is true, but it should be clarified that the main role of mTORC1 is the switch between autophagy and protein biosynthesis. However, only one sentence is devoted to autophagy in this description. I believe that this description should be expanded and, moreover, more attention should be paid to it, as it is known that autophagy itself also regulates the development, differentiation and activation of T cells. See for example: (1) Bronietzki et al (2015) Immunol Cell Biol 93(1):25-34; (2) Merkley et al (2018) Front Immunol 9:2914.
We have clarified the role of mTORC1 in promoting anabolic vs catabolic process (eg autophagy), please see lines 97-99.
We have added a new section on the role of autophagy as suggested by the reviewer. Please see new section 5.1.5.
3) The abstract is incorrectly written. Too much space has been devoted to the description of the selection of T lymphocytes and their formation, and too little to the role of mTOR in regulating the differentiation of T lymphocytes and thymocytes. The description of the formation of the immune system cells themselves (lines 12-21) should be shortened and the description of a more precise role of mTOR and its regulation in the described processes of the selection of these cells extended, which is the main subject of this manuscript (lines 22-25). The abstract should be a summary of the article and not an introduction to the main topic.
We have now improved the abstract as suggested by the reviewer.
Minor comments:
4) The authors at the end of some chapters provide a plan of descriptions in the following chapters (lines 47-55; 323-325; 431-437). I don't think such plans are needed. This distracts the reader from the continuity of the text and generates unnecessarily extra space in the already long manuscript.
We have removed these lines as suggested.
5) Please check the spacing. Throughout a manuscript, it seems to me that there is often a double space before starting a new sentence.
We have corrected this as suggested.
6) There is a lack of complete affiliation data.
We have corrected this now.